# Tuning organic crystal chirality by the molar masses of tailored polymeric additives

Xichong Ye [1], Bowen Li[1], Zhaoxu Wang[1], Jing Li[1], Jie Zhang[1] & Xinhua Wan [1✉]

Hierarchically ordered chiral crystals have attracted intense research efforts for their huge potential in optical devices, asymmetric catalysis and pharmaceutical crystal engineering. Major barriers to the application have been the use of costly enantiomerically pure building blocks and the difficulty in precise control of chirality transfer from molecular to macroscopic level. Herein, we describe a strategy that offers not only the preferred formation of one enantiomorph from racemic solution but also the subsequent enantiomer-specific oriented attachment of this enantiomorph by balancing stereoselective and non-stereoselective interactions. It is demonstrated by on-demand switching the sign of fan-shaped crystal aggregates and the configuration of their components only by changing the molar mass of tailored polymeric additives. Owing to the simplicity and wide scope of application, this methodology opens an immediate opportunity for facile and efficient fabrication of one-handed macroscopic aggregates of homochiral organic crystals from racemic starting materials.

[1] Beijing National Laboratory for Molecular Sciences, Key Laboratory of Polymer Chemistry and Physics of Ministry of Education, College of Chemistry and Molecular Engineering, Peking University, Beijing 100871, China. ✉email: xhwan@pku.edu.cn

Hierarchically ordered chiral assemblies like biominerals, scales, plant fibers are abundant in nature and play indispensable roles in the whole process of life[1–3]. They have inspired the development of artificial systems[4] showing huge potential in chiral resolution[5], asymmetric catalysis[6,7], optical devices[8–11], chiral sensors[12,13], pharmaceutical crystal engineering[14], and so forth. The formation of these chiral biostructures relies on the accurate recognition and screening of optically active compounds as well as the precise hierarchical self-assembling of building blocks that are selected[15]. Even though unnatural chiral compounds (e.g., D-amino acids and L-saccharides) are accidently ingested by taking artificial foods and medicines[16,17], they would hardly be incorporated into biomacromolecules and the corresponding assemblies[18]. In contrast, to form optically active structures, man-made strategies currently available have to either use enantiomerically pure components (Fig. 1, route i)[8,11,19–23] or separate the mixture of assemblies with mirror-images formed from racemic compounds through a self-sorting process (Fig. 1, route ii)[5,24,25], due to lacking the ability of auto-screening unfitted building blocks. Stereoselective generation of hierarchically ordered chiral crystals of a single enantiomer from a racemic solution in one-step would greatly improve the production efficacy (Fig. 1, route iii) but has never been realized.

Oriented attachment (OA) is an efficient method to generate hierarchically ordered crystals[2,26–29], which is a dynamically preferential pathway to reduce surface energy of small particles by ordered attaching solid phase blocks to the growing surfaces, especially in high concentration solutions[30]. Stabilized crystalline building blocks with high shape anisotropy are essential to this process[31,32], which can be achieved by using additives[29,33] or particular ripening conditions[5,24]. *Rac*-threonine has been reported to form the mixture of right- and left-handed crystal clusters through enantiomer-specific oriented attachment (ESOA)[5], and only 85% ee can be achieved for each of them. By using chiral additives, imperfect OA occurs and chiral mesocrystals can be generated. For examples, small chiral molecules are often used as additives to fabricate chiral mesocrystals by being incorporated between two crystalline plates and inducing twisted allignment of these subunits[22,23,25,34]. Most of the buliding blocks are inorganic materials, studies on pure organic chiral mesocrystals are rarely reported. The attempt to obtain chiral meso-structure from *rac*-alanine has been conducted by using chiral double-hydrophilic block copolymers[33]. It was found that although no chiral resolution was achieved, the polymeric additives could stabilize the crystalline subunits and thus changed the crystal habit. The additives selectively attach on the specific crystal faces and slow down their growth[35]. This rule has long been applied for stereoselective crystallization of conglomerates[36–40]. Chiral additives delay the nucleation and growth of one enantiomorph with the same configuration through stereoselective adsorption. In fact, the attachment of the additives with opposite configuration can also change the morphology of the crystal owing to the non-stereospecific interactions between the additive molecules and specific faces[37,41]. However, the non-stereoselective interactions have been overlooked in building hierarchically ordered chiral organic crystals. We posit that macroscopic homochiral crystal aggregates of a single enantiomer could be generated from a racemic solution by virtue of synergistic combination of stereoselective and non-stereoselective interactions.

Herein, we describe a strategy that enables the preferred formation of one enantiomorph directly from racemic solutions as well as their in-situ OA to fabricate macroscopic chiral crystal

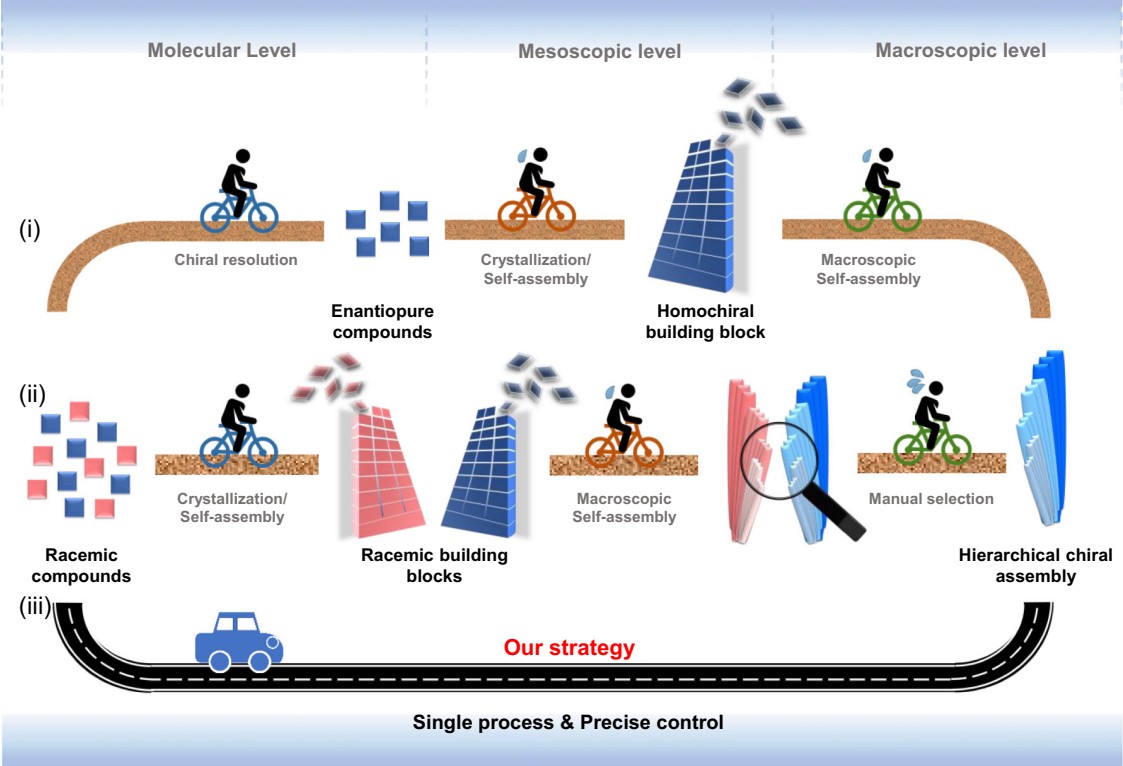

**Fig. 1 Scheme of concepts.** Route i: Enantipure molecules are obtained through chiral resolution, form enantipure crystals and one-handed crystal aggregates. Route ii: Racemic compounds form a pair of enantiomorphs through a self-sorting process, which then aggregate into chiral crystal aggregates with opposite sense, separately. Manual selection is needed to obtain optically active structures. Route iii: In this work, the hierarchical chiral structures are directly generated from racemic solutions in a single process.

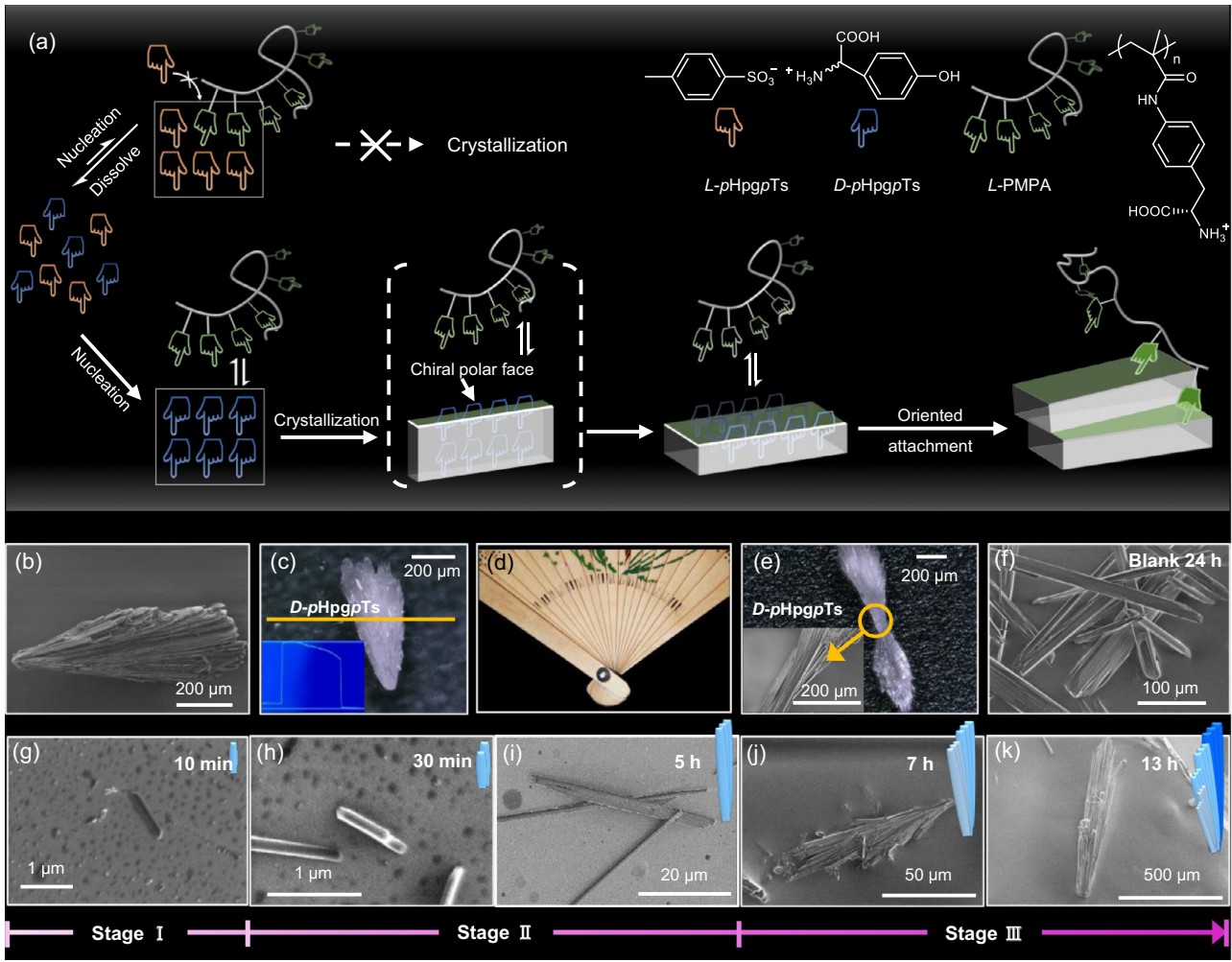

**Fig. 2 The formation of hierarchically ordered chiral crystals from racemic solution. a** The scheme of our strategy: *L*-PMPA with suitable chain length completely inhibits the nucleation and growth of *L-p*Hpg*p*Ts and influences the crystal habit and oriented attachment of *D-p*Hpg*p*Ts through adsorption-desorption process. **b** Optical micrograph of a fan-shaped crystal aggregate of *D-p*Hpg*p*Ts when 1.5 ₍wt₎% P5 was used as the additive. **c** Picture of one P-type fan-shaped crystal of *D-p*Hpg*p*Ts taken by an optical microscope with large depth-of-field. (inserted picture: the height map of the crystal alone the yellow line). **d** Picture of a left-handed (P-type) folding fan. **e** Optical micrograph of one *D*-crystal with two-blade propeller morphology (inserted picture: the enlarged SEM image of the circled spot). **f** The typical SEM images of needle-like crystals of *D/L-p*Hpg*p*Ts obtained at 24 h when no additive was added. **g–k** The typical SEM images of the obtained crystals at 10 min to 13 h when 1.5 ₍wt₎% P5 were used as additives. And P-type fan-shaped crystals of *D-p*Hpg*p*Ts were the only species under this condition.

aggregates, only aided by the tailored polymeric additives with various molar masses. When a *L*-polymer with a medium molar mass is used, Plus (P)-type fan-shaped crystal aggregates of *D-p*-hydroxyphenylglycine *p*-toluenesulfonate (*D-p*Hpg*p*Ts) are obtained (Fig. 2a). Changing the molar mass of the polymer can switch the chirality of the hierarchically ordered structures thus formed on both molecular and supramolecular levels (Supplementary Fig. 1).

## Results

**Formation of homochiral crystal aggregates**. Nine narrowly distributed poly[*p*-methacrylamido *tert*-butoxycarbonyl-*L*-phenylalanine] (*L*-PMPABoc)s with number-average molar masses ($M_n$s) ranging from 5700 to 30,100 Da were synthesized by reversible addition-fragmentation chain transfer radical polymerization, followed by purification using preparative gel permeation chromatography. The corresponding deprotected poly[*p*-methacrylamido-*L*-phenylalanine]s (*L*-PMPAs) were named as P1–P9 (Table S1, Supplementary Fig. 2). The polymer with a moderate molar mass, i.e. P5 ($M_n = 13,500$ Da,

PDI = 1.10), was first used as the additive to verify our ideas. *rac-p*Hpg*p*Ts was chosen as a model substrate. Into a supersaturated solution of *rac-p*Hpg*p*Ts (2.5 g of *rac-p*Hpg*p*Ts in 10 mL of 0.5 M *p*-toluenesulfonic acid aqueous solution), that was prepared and filtrated at 60 °C, 1.5 ₍wt₎% (polymer/*rac-p*Hpg*p*Ts) of P5 was added. The crystallization was allowed to take place at 25 °C. After a period of time, the formed crystals were collected by filtration and washed with cold acetone. After drying under vacuum, their morphologies were examined by optical and electron microscopes (Supplementary Methods).

Fan-shaped crystal aggregates were exclusive species when P5 was used as the additive (Fig. 2b), which was quite different from randomly distributed needle-like crystals formed in the absence of additive (Fig. 2f, Supplementary Fig. 4). The height map of fan-shaped crystals given by optical microscope with a large depth-of-field (Fig. 2c, inserted picture) displayed a gradually rising profile from the right edge to the left edge, indicating a macroscopic P-type twisting sense which is just like a left-handed folding fan (Fig. 2d). Occasionally, crystals with two-blade propeller morphology were observed (Fig. 2e), where the twisting sense

was more obvious. Further investigation showed that the fan-shaped structures were composed of small platelets (Supplementary Fig. 5). Chiral HPLC tests showed that the P-type fan-shaped crystals composed of $D$-$p$Hpg$p$Ts exclusively (Supplementary Fig. 6a), while needle-like crystals obtained in blank control experiment were the mixtures of enantiomorphs consisting of $D$- and $L$-$p$Hpg$p$Ts, separately (Supplementary Fig. 7). Crystallization was also carried out in optically pure $D$-$p$Hpg$p$Ts solution. Only needle-like crystals were obtained in the absence of additives. Whereas, fan-shaped crystals were produced when P5 was added (Supplementary Fig. 8). These results proved that the emergence of fan-shaped crystals was attributed to the presence of chiral additives but not merely the chirality of amino acid itself.

Crystals obtained at different time were collected and examined by SEM and circular dichroism spectra (CD) to investigate the evolution process. Three distinct stages could be identified in the crystallization process (Fig. 2g–k, Supplementary Fig. 9). In the first 10 min (stage I), small round particles were dominated, although some tiny crystals with a length around 1 μm could also be found occasionally (Fig. 2g). The CD signal grows linearly until it reaches a plateau (Supplementary Fig. 10a, b). From 30 min to 5 h (stage II), more tiny crystalline platelets appeared. Meanwhile, the stacking of two or three platelets at a twisting angle of $-1 \sim -2°$ (counterclockwise) was also present (Fig. 2h, i) with the CD signal increasing further (Supplementary Fig. 10b). During this period, the crystals grew up to a length of 40 μm and the prototype of fan-shaped crystals began to emerge. From 7 to 13 h (stage III), fan-shaped crystals were observed. These crystals had a length around 600 μm and a width around 160 μm (Fig. 2j, k). Moreover, several steps from the right edge to left edge were observed (Fig. 2k and Supplementary Fig. 11). At this time, a well-developed cotton effect appears in CD spectrum, indicating the emergence of high-level chirality (Supplementary Fig. 10c). These results suggested that the final fan-shaped crystals were gradually assembled by the platelet sub-structures that formed in the earlier stages through imperfect OA process.

To further confirm this assumption, the self-assembling of pre-prepared $D$-$p$Hpg$p$Ts crystals was conducted. Needle-like $D$-crystals were prepared by recrystallization of $D$-$p$Hpg$p$Ts without adding any additive and ground lightly to smaller sizes. They were then added into the saturated solution of $D$-$p$Hpg$p$Ts at room temperature followed by adding 1.5 $_{wt}$% P5. After a period of ripening, crystals were filtrated out and observed by SEM. Morphological reconstruction was examined (Supplementary Fig. 12): in the first 4 h, the surfaces of needle-like crystals were partially dissolved and the crystal edge became blurred; From 4 h to 24 h, platelets with new developed faces and aggregates of two or three platelets were observed. Ordered arranged crystal aggregates formed after 108 h.

**Chirality tuning**. The twisting sense of the fan-shaped crystal aggregates and the configuration of constituting amino acid molecules were highly dependent on the $M_n$s of the polymeric additives at a fixed concentration of 1.5 $_{wt}$% (Fig. 3a). Although P5 yielded exclusively P-type fan-shaped crystal aggregates consisting of $D$-$p$Hpg$p$Ts, predominant Minus (M)-type fan-shaped crystal aggregates consisting of $L$-$p$Hpg$p$Ts along with some needle-like crystals consisting of $D$-$a$. $a$. were obtained when $p$-methacrylamido $tert$-butoxycarbonyl-$L$-phenylalanine ($L$-monomer) or $L$-PMPAs with relative low $M_n$s (i.e. P1-P3) were used (Fig. 3d, Supplementary Figs. 13–16). The formation of M-type fan-shaped crystals showed a similar but faster evolution process (Supplementary Fig. 17). And the neighboring platelets twisted around 1~2° (clockwise, Fig. 6b). In the case that $L$-PMPAs with

larger $M_n$ (P8-P9) were used, roughly equal amount of M- and P-type fan-shaped crystal aggregates were obtained (Fig. 3d). Moreover, CD spectra of M- and P-type crystals showed opposite cotton effect at the absorption range which is consistent with the HPLC results (Supplementary Fig. 18).

The chirality bias and morphologies of the crystal aggregates were also dependent on the additive concentration. Although 1.5 $_{wt}$% of $L$-monomer resulted in M-type fan-shaped aggregates of $L$-$p$Hpg$p$Ts crystals, raising the amount of $L$-monomer to 10 $_{wt}$% switched the twisting sense of crystal aggregates and the configuration of constituting amino acid (Fig. 3b). Besides, increasing the amount of P9 (up to 5.0 $_{wt}$%) raised the ee% value of the resultant crystals (Fig. 3c) and P-type fan-shaped crystals became the dominated products.

Overall, the macroscopic and molecular chirality can be tuned at the same time by changing the molar masses of polymeric additives (Supplementary Table 2), which has never been reported in the formation of hierarchically ordered crystals. In traditional methods, chirality tuning of hierarchical structures is usually achieved by changing the chirality of chiral components[19–21,25]. It is also reported that the apparent chirality can be switched by extending aging time[22,42], as the building blocks have a staggered orientation that leads to an overall helical structure in one handedness, while the complex aggregates formed by these structures show helicity in the other handedness.

**Distribution of polymeric additives**. The distribution of the polymers in crystals was investigated. Fluorescein labeled polymer ($L$-PMPA(Flu)) with a similar $M_n$ as P5 was prepared (see Supplementary Method and supplementary Table 1)[43]. And the crystallization was carried out under the same condition mentioned above. The crystals were collected to test their emission. The contents of polymers in the crystals were calculated to be 0.2 $_{wt}$% according to the standard curve (Fig. 4a and Supplementary Fig. 19), indicating a high purity of crystals. Furthermore, the fluorescent images showed that the small amount of polymers selectively gathered on the long-axis direction of each single crystalline unit (Fig. 4b, c). To investigate whether the polymers exist between two crystalline building blocks, two models were discussed. If polymers existed between two neighboring platelets, the fluorescent image of the cross-section in x direction should be a periodic pattern of light and shade, and the cross-section in y direction should has a similar periodic fluorescent image (Supplementary Fig. 20). If polymers weren't involved between two platelets, the fluorescent image of the cross-section in x or y direction should be a step-like pattern (Fig. 4e). Actually, in our case, no periodic fluorescent spots along the z axis in 3D fluorescent images (Fig. 4d), indicating there is no polymer-crystal-polymer sandwich structures. Besides, the fluorescent spots on the x-z plane gradually rose from +x to −x and the fluorescent spots on the y-z plane gradually rose from +y to −y (yellow arrows in Fig. 4d), which was consistent with the step-like fluorescent pattern shown in Fig. 4e. When the crystals were progressively dissolved by cold water, the intensity of fluorescence emission dropped dramatically (Supplementary Fig. 21), indicating again that the polymers only existed on the surface of the whole crystal and no polymer existed between two subunits.

Double-hydrophilic block copolymers has been reported to act as a glue to adhere tightly to the polar faces and join the tiny crystals together. However, in our case, the side-chains of the polymer are hydrophilic and the mainchain is hydrophobic. When the side chains attached onto the polar face of one crystal platelet, the outside mainchains could not attach onto another one to join them together. Besides, due to the relatively weak non-

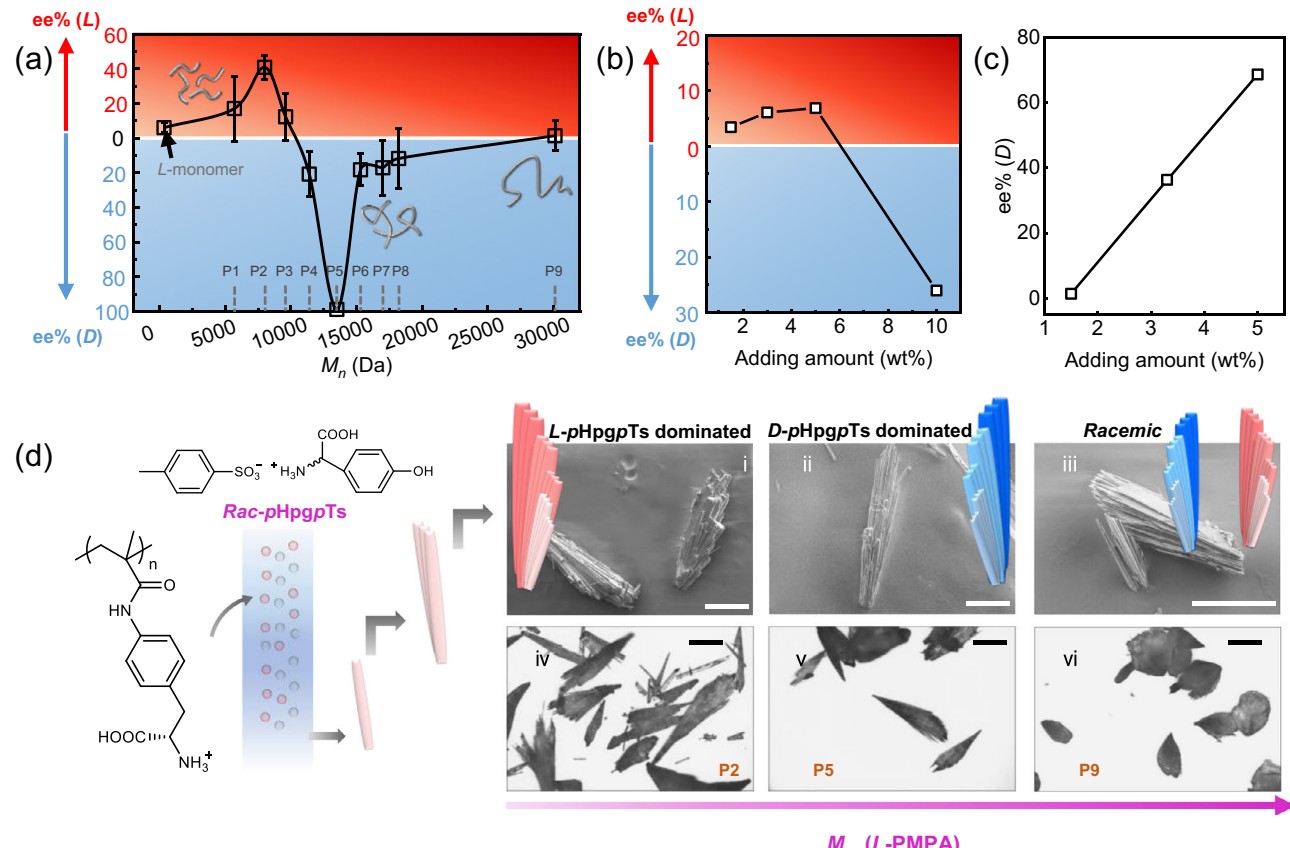

**Fig. 3 Chirality tuning by using polymeric additives with various molar masses. a** The ee% values of the obtained crystals by using *L*-monomer or *L*-polymers with various molar masses as additives. Error bars are standard errors calculated from five independent experiments. **b** The ee% values of the obtained crystals by using different amount of *L*-monomers. **c** The ee% values of the obtained crystals by using different amount of P9. **d** The hierarchically ordered crystals were generated from racemic solutions and their inner and apparent chirality can be tuned by changing additives' molar masses: (iv–vi) Optical micrographs of the crystals obtained by using polymers with different $M_n$s (8000, 13500 and 30100 Da) as additives. (i–iii) the corresponding partially enlarged SEM images, showing the detailed morphologies of the fan-shaped crystals. Scale bar: 200 μm.

stereospecific interactions, it is highly possible for these polymers to leave through adsorption-desorption dynamic processes. When the OA happened, the polymers desorbed from the adhering area to reduce surface energy and resided only on the exposed surface.

**Computational simulations**. To investigate how the polymers influence the crystal habit, computational simulations were conducted. Single-crystal of *D-pHpgp*Ts belongs to orthorhombic system ($a = 5.45$ Å, $b = 14.41$ Å, $c = 19.57$ Å) with $P2_12_12_1$ symmetry (Fig. 5a and Supplementary Fig. 22c, d). The crystal habit was observed by SEM and simulated by corrected attachment energy (AE) model based on the unit cell (Supplementary Theory). The crystal habit in pure water was predicted to be a rod-like prism and elongated along the a direction (Fig. 5b), which was identical to the actual single-crystal (Fig. 5a). The {0 1 1}, {0 0 2}, {1 0 1} and {1 1 0} faces were exposed, and their surface chemistry and topography could be revealed (Supplementary Fig. 23): {0 1 1} faces were the most polar faces, which were along the polar b axis where -$NH_3^+$, -COOH and -$SO_3^-$ groups face straight outward (Fig. 5d green layer); -COOH and -$SO_3^-$ groups also exposed on the {1 0 1} faces which were along a axis (Fig. 5e green layer). The {0 1 1} faces were morphologically the most important whose total facet area was 61.8%, while total facet areas of {0 0 2}, {1 0 1} and {1 1 0} were 25.8%, 9.3% and 3.1% (Fig. 5f). As high concentration monomer and moderate molar mass polymers had the same effect (Fig. 3b, Supplementary Fig. 24), a solvent layer with 185 water

molecules and 15 *L*-monomers (7.5 mol%) was built to mimic these systems (Fig. 5d, e and Supplementary Fig. 26). Compared with that in pure water, higher interaction energy ($E_{int}$) was given between additives and {0 1 1} and {1 0 1} faces (Fig. 5g), leading to a larger $E_s$ ($E_s = E_{int} \times S$, where $S$ is the accessible solvent area per unit facet area, see detail in Supplementary Theory) and a lower $E_{att}'$ (corrected attachment energy, $E_{att}' = E_{att} - E_s$) (Fig. 5h, i and Supplementary Table 9). According to the corrected AE model, the $E_{att}'$ was proportional with the relative growth rates ($R(hkl)'$). Thus, a lower relative growth rate and a higher morphological importance of {0 1 1} and {1 0 1} was obtained. As for {0 0 2} faces, the $E_{int}$ and $E_s$ were lower than that in pure water, and the $E_{att}'$ became higher, leading to a much less total facet area. As a result, a flat prismatic crystal with a smaller aspect ratio was obtained (Fig. 5c) in the additives' solution, the {1 1 0} faces disappeared, {0 1 1} and {1 0 1} became the dominated faces with total facet areas of 77.51% and 20.13% (Fig. 5f).

Corrected AE model with a solvent layer of 190 water molecules and 10 *L*-monomers (5 mol%) was built to mimic the system when less monomers or short chain polymers were added. According to the simulations, crystal habits of *D*- and *L*-crystals in this solution turned out to be different: For *L*-crystals, the {1 1 0} faces disappeared, and the total facet area of chiral polar faces {0 1 1} increased from 56.92% to 74.09%, and {0 0 2} facet area decreased from 27.10% to 5.03% (Supplementary Fig. 32). The crystal habit turned out to be a flat prism with smaller aspect ratio. As for *D*-crystals, the total facet area of {0 1 1} and {0 0 2}

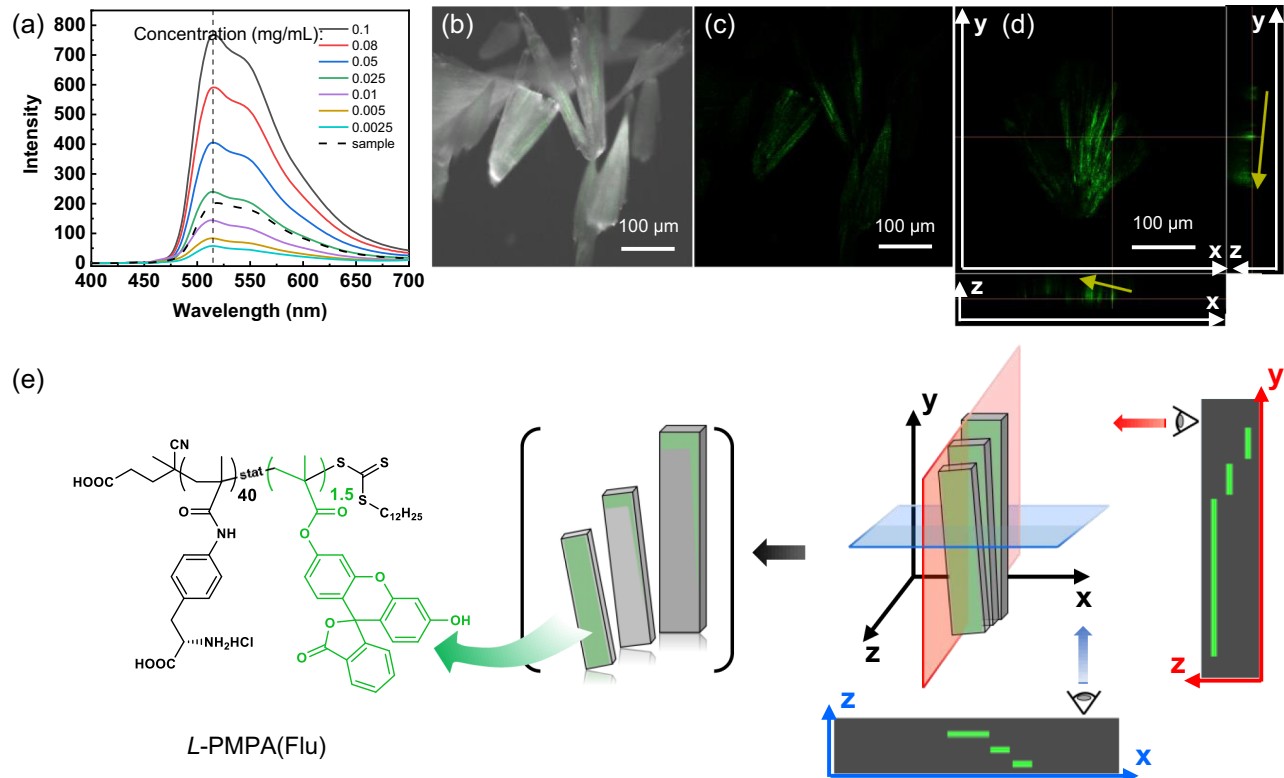

**Fig. 4 The residual amount of *L*-PMPA(Flu) and their distribution in the crystal aggregates of *D*-*p*Hpg*p*Ts. a** Fluorescence emission spectrum of *L*-PMPA(Flu) aqueous solutions with different concentration (0.0025–0.1 mg/mL), the dashed line is the curve of crystal sample (100 mg crystals dissovled in 10 mL H₂O). **b** Merged picture of optical micrograph and fluorescence image of fan-shaped crystal aggregates of *D*-*p*Hpg*p*Ts when 1.5 wt% *L*-PMPA(Flu) was used as the additive. **c** The corresponding 2D confocal fluorescence micrographs. **d** The 3D confocal fluorescence profile of one fan-shaped *D*-crystal. **e** The model for the situation that no polymers exist between two crystalline platelets but only on the surface of the whole fan-shaped crystal, and the corresponding fluorescein patterns on the cross-sections (blue plane: cross-section in y direction; red plane: cross-section in x direction). The crystal habit was simplified to a rectangular block for better classification. The molecular structure of *L*-PMPA(Flu) is illustrated on the left.

didn't change too much (Supplementary Tables 7 and 8), which were still prismatic crystals with large aspect ratio (Supplementary Fig. 27).

In order to study the relationship between molecular chirality and macroscopic chirality of fan-shaped crystals, the attachment angle of two morphological changed *D*- or *L*-*p*Hpg*p*Ts crystals were calculated by building a model box containing a main crystal and a smaller one. The two crystalline layers were attached by (0 1 1) and (0 −1 −1) faces, and the total energy was calculated when the upper layer was rotated from −6° ~ +6° (Supplementary Fig. 34). It turned out that the lowest total energy for *D*-crystals appears at −1° (Supplementary Fig. 35a), while the lowest total energy for *L*-crystals appears at +1° (Supplementary Fig. 35b), which showed a good agreement with the experimental results. As the electrostatic interaction is dominant in the total energy (Supplementary Fig. 35c), the possible reason may be that the ion pairs on the interface can be perfectly matched only when the two layers rotate a small angle.

**Mechanisms**. On the basis of these results, it is possible to elucidate the interactions between the additives and the crystals as well as the formation of hierarchically ordered chiral crystal aggregates. *D/L*-*p*Hpg*p*Ts forms polar crystals and displays well-developed {0 1 1} faces, where -NH₃⁺, -COOH and -SO₃⁻ groups face straight outward, and {1 0 1} faces along a axis where -COOH and -SO₃⁻ groups exposed. *L*-PMPA with phenylalanine side groups can attach on these polar faces by multiple hydrogen bonds, and the relative strength depends on molar masses of additives at a fixed concentration[44–46].

*L*-PMPA with an optimized chain length (i.e., P5) interacts stereoselectively and strongly with the clusters of the solute molecules with same configuration (i.e., *L*-*p*Hpg*p*Ts), that are smaller than the critical nucleus size, and efficiently inhibits the crystal nucleation and growth. In the meantime the *L*-PMPA molecules can also bind on the specific surfaces of *D*-*p*Hpg*p*Ts crystals through non-stereoselective interactions. Although such interactions are relatively weak and cannot stop the crystallization of *D*-*p*Hpg*p*Ts, the crystal growth perpendicular to {0 1 1} and {1 0 1} faces is delayed. With the enlarged area of theses faces, the *D*-*p*Hpg*p*Ts crystals are oriented along the a direction and stacked together along the b direction to reducing their high surface energy (Supplementary Fig. 28). Powder XRD showed that the intensity of (0 1 1) peak was increased and that of (0 0 2) peak was decreased when 1.5 wt% of P5 was used as the additive (Supplementary Fig. 33). The {1 0 1} faces also showed high polarity, but the facet area was much smaller than that of {0 1 1} faces, making it unlikely to attach from this direction.

*L*-monomer and short chain *L*-PMPA interact stereoslectively with the clusters of *L*-*p*Hpg*p*Ts. However, the strength is weak and cannot inhibit the crystallization of this enantiomer due to the low molar masses of additives. Whereas, the fast adsorption-desorption equilibrium on {0 1 1} and {1 0 1} faces change the crystal habit of *L*-*p*Hpg*p*Ts and favor the OA process. Also due to the weak non-stereoselective interactions exerted by *L*-monomer and short chain *L*-PMPA, the crystal nucleation and growth of *D*-*p*Hpg*p*Ts are not obviously affected, and needle-like crystals are obtained. When long chain *L*-PMPA were used, the strong non-stereoselective interactions with *D*-*p*Hpg*p*Ts change its crystal habit and induce P-type

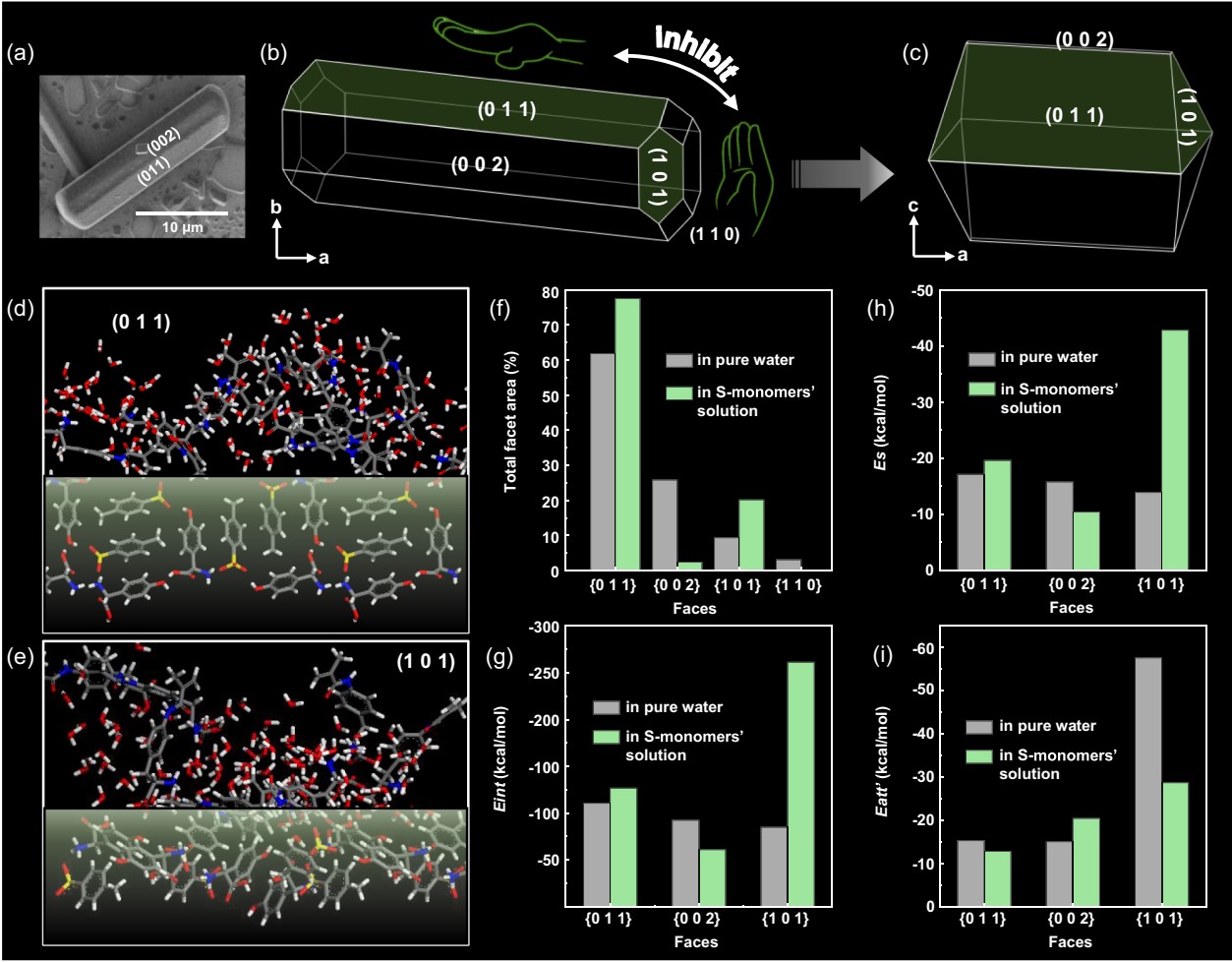

**Fig. 5 Mechanism for morphological changes of *D-p*Hpg*p*Ts. a** SEM image of a *D-p*Hpg*p*Ts single-crystal obtained from dilute aqueous solution by slow evaporation. **b** The simulated crystal habit of *D-p*Hpg*p*Ts in pure water, the green faces are more likely to be attached by additives. **c** The simulated crystal habit of *D-p*Hpg*p*Ts in 7.5 mol% *L*-monomer aqueous solution. **d** Surface structure of (0 1 1) face (green layer) with amino and carboxyl groups facing straight outward. After the molecular dynamics (MD) simulation, the molecules of $H_2O$ and *L*-monomers adsorbed on the surface. **e** Surface structure of (1 0 1) face (green layer) and the absorbed solution molecules after MD simulation. **f** The total facet area (%) of different faces in pure water and 7.5 mol% *L*-monomer's aqueous solution. **g** $E_{int}$ of different faces in pure water and 7.5 mol% *L*-monomer's aqueous solution. **h** $Es$ of different faces in pure water and 7.5 mol% *L*-monomer's aqueous solution. **i** $E_{att}'$ of different faces in pure water and 7.5 mol% *L*-monomer's aqueous solution.

fan-shaped crystal aggregates. On the other hand, the stereoselective interactions with *L-p*Hpg*p*Ts cause uneven local concentration of the additives[44–46]. The crystal nucleation and growth of this enantiomer are not inhibited, but the crystal habit is changed. As a result, either M- or P-type fan-shaped crystals composed of *L*- and *D-p*Hpg*p*Ts, respectively, are obtained.

Raising the concentrations of additives helps the adsorption on *L*-clusters through stereoselective interactions and the crystals of *D-p*Hpg*p*Ts through non-stereoselective interactions. The crystallization of *L-p*Hpg*p*Ts is thus inhibited while *P*-type fan-shaped crystal aggregates of *D-p*Hpg*p*Ts are obtained.

**Applicability**. To explore the scope of this method, the crystallizations of racemic threonine (*rac*-Thr), allo-threonine (*rac*-aThr), aspartic acid (*rac*-Asp)[47] and aspartic acid copper complex (*rac*-Asp₂Cu)[48,49] were carried out in the presence of Poly(*N*⁶-methacryloyl-*L*-lysine) (*L*-PMAL)[37,45,46,50]. In all cases, *D*-crystals with over 90 ee% were obtained. More importantly, crystal aggregates were always observed when *L*-PMAL was used as additive (Fig. 6 and Supplementary Figs. 36 and 37). Specifically, *D*-Asp₂Cu crystals (92.3 ee%) with P-type helical morphology

were generated in 6 h when 5 wt% *L*-PMAL ($M_n$ = 7500 Da) was added (Fig. 6b, c and Supplementary Fig. 38), whereas the strands of wirelike crystals without chiral morphology were dominant in the absence of additive (Fig. 6a). Besides, when using monomer as additive, *L*-Asp₂Cu with 26.5 ee% can be obtained, and M-type helical crystals can be observed (Supplementary Fig. 39). *L*-PMAL along with ammonium formate were used to control the crystallization of *rac*-Asp, M-type helicoid *D*-Asp (90.5 ee%) was obtained, whereas only disordered granular crystals were observed in blank control (Fig. 6d–f).

## Discussion

We have reported a strategy to prepare chiral crystal aggregates of a single enantiomer from a racemic solution by using tailored polymeric additives. The chirality on both molecular and macroscopic levels can be switched by changing the molar masses of additives. Particularly, *P*-type fan-shaped crystal aggregates of *D-p*Hpg*p*Ts are obtained when *L*-PMPA with an optimized molar mass is used. This unique and efficient chiral regulation method is realized by balancing the strength of stereoselective and non-stereoselective interactions between *L*-additive and *D/L-p*Hpg*p*Ts

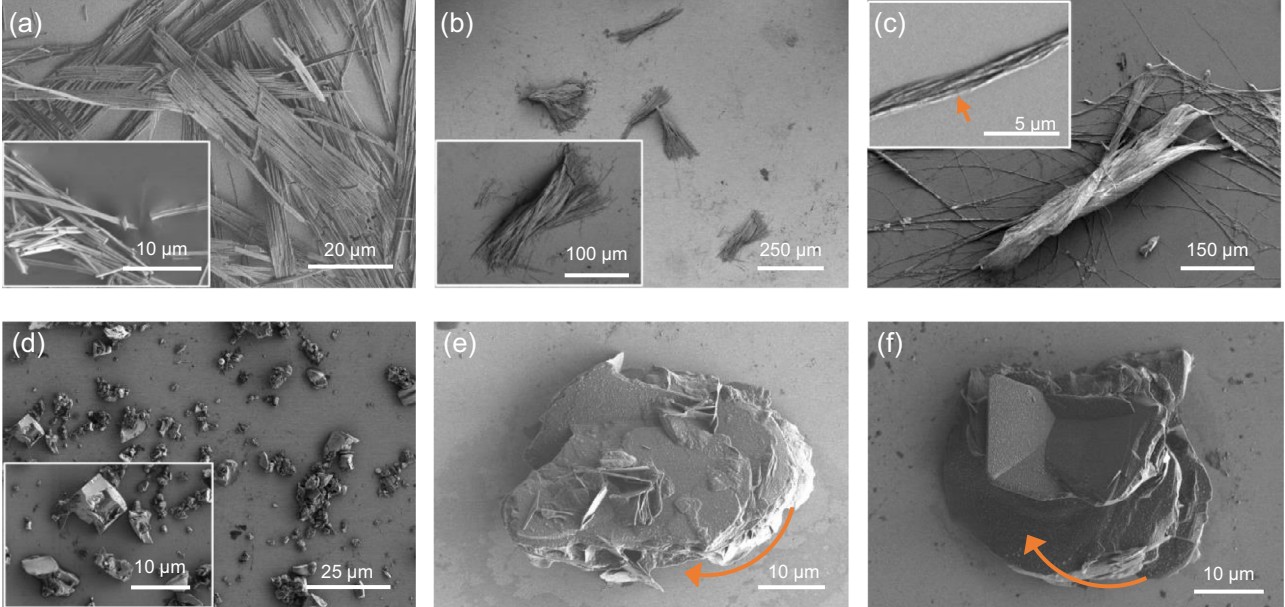

**Fig. 6 The typical morphologies of Asp$_2$Cu and Asp crystals from their racemic solutions with or without additives. a** The typical morphology of *rac*-Asp$_2$Cu when no additive was added (insert picture: enlarged picture of the strands of wirelike crystals). **b** The typical morphology of *D*-Asp$_2$Cu when 5 $_{wt}$% *L*-PMAL was added into the solution of *rac*-Asp$_2$Cu (insert picture: enlarged picture of the P-type helical crystal). **c** A P-type helical crystal of *D*-Asp$_2$Cu composed of three strands of crystals (insert picture: enlarged picture of one strand of crystals, from which twisted alignment of wirelike crystals in right-handedness can been seen). **d** The typical morphology of *rac*-Asp when no additive was added (insert picture: enlarged picture of the granular crystals). **e**, **f** The typical morphologies of *D*-Asp when 1.5$_{wt}$% *L*-PMAL (*L*-PMAL:*rac*-Asp, wt/wt) and 3 e.q. ammonium formate (NH$_4$HCO$_2$:*rac*-Asp, mol/mol) were added into the solution of *rac*-Asp. The red arrows show M-type helical sense.

nucleus and crystals. The application of this method has successfully been extended to other conglomerate forming racemates (i.e. Thr, aThr, Asp, and Asp$_2$Cu) by using *L*-PMAL as the additive. It has considerably simplified the fabrication protocol of hierarchical chiral structures from racemic compounds. Low additive dosage and high purity of the crystalline product also make it suitable for large scale production. This work would inspire further research to increase our fundamental understanding in accurate chiral discrimination and cross-scale, multi-level transmission, and expand the scope of accessible building blocks. We envision the potential wide applications of this strategy in pharmaceutical crystal engineering, organic chiral micro-/nano-laser, and asymmetric catalysis.

## Data availability

The data that support the findings of this study are available from the corresponding author upon request. The Crystallographic data generated in this study have been deposited in the CCDC database under deposition numbers 2106333 and 2106336 [https://www.ccdc.cam.ac.uk/solutions/csd-core/components/csd/]. Source data are provided with this paper.

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

## Acknowledgements
We gratefully acknowledge the financial supports from the National Natural Science Foundation of China (Nos. 51833001; 21674002; 21905003) (X.H.W.), and the financial supports from the China Postdoctoral Science Foundation (No. 2019M660002; 2020T130011) (X.C.Y.). We thank Prof. Jiaxi Cui for his valuable advice and suggestions on revision of the manuscript.

## Author contributions
X.H.W. supervised the research. X.C.Y. and X.H.W. contributed to the conception, design of experiments, drafting and critical revision of the manuscript; X.C.Y., B.W.L., Z.X.W. and J.L. contributed to synthesis, analysis and data collection; J.Z. contributed to discussion of experiment results and revision of the manuscript.

## Competing interests
The authors declare no competing interests.
