## [Peer Review File · Nature Communications]

REVIEWER COMMENTS

Reviewer #1 (Remarks to the Author):

This manuscript describes the isolation-crystallization of one enantiomer from a racemic mixture using a polymeric material that contain side groups of one of the enantiomers. The main finding of this article is that the amount added and the molecular weight of the polymer dictate the crystallization of one of the enantiomers. Thus one polymer additive can provide the separation of a racemic mixture by inducing crystal formation of one enantiomer.

This is a nice and detailed work that demonstrate one specific system tailored for the separation of a certain racemic mixture. It seems that for each racemic mixture, a polymeric version should be tailored and investigated to obtain selective crystallization.

The process described is not trivial and requires tailoring polymers of certain structure and molecular weights that may fit one set of racemic mixture and does not provide a general process where one is adding a magic molecule that fits many racemic mixtures, this would justify special attention. This work is specific and not of a general solution but a finding that is of interest for experts in the isolation procedures. Moreover, recent developments in the isolation of racemic mixtures using novel methods have been recently published.

Inducing crystallization of one enantiomer by adding a chiral agent is well known, this work is within this scope and provides a farther example of these efforts. This work is not a break through that justify publication in this journal. It should be directed to a specific journal that deal with racemic separations.

More specifically, this article is hard to read, the illustrations are difficult to follow as well as the details of the procedures.

In summary, this article is not recommended for publication in this journal, it should be forwarded to a specific journal in this field.

Reviewer #2 (Remarks to the Author):

In this manuscript, Ye et al. reported a facile and effective route to achieve the enantioselective crystallization and assembly of hierarchically ordered chiral crystals from racemic molecules by introducing polymeric additives. By regulating the molar masses and amount of polymeric additives, the chirality of the molecular crystal aggregates can be modulated, yielding either P- or M-type fan-shaped crystal aggregates of controllable ee% values. In particular, they reported enantiopure second-order assembly using polymer of P5. The accessibility of enantiopure macroscopic organic crystals from racemic starting materials without further manual purification is of great significance for chirality-related applications. Furthermore, combining theoretical calculations and experimental results, the authors also revealed the underlying mechanism that enables such chirality tenability.

Overall, this work is interesting and can be considered for publication in Nature Communications after following comments are fully addressed.

1. The twisting direction of the resulting crystal aggregates is relatively difficult to resolve, the authors may provide eye-guided SEM images of higher resolution.

2. The numbers of samples counted in Supplementary Figure 14 for statistical ee% values seem far from enough. Would the circular dichroism spectrum manifest their enantioselectivity differences in Figure 3(a)? The authors are encouraged to present CD spectra or g-factor spectra of these crystal aggregates to evaluate the ensemble chirality differences using different polymeric additives.

3. As shown in Figure 3, both the Mn and adding amount of polymeric additives show influences on the chirality of hierarchical assembly. Thus, is it possible to achieve both enantiopure (ee value of 100%) P- and M-type assembly by synergistically modulating Mn and adding amount of polymeric additives?

4. As described in the manuscript, the synergistic combination of stereoselective and non-stereoselective interactions of the polymeric additives results in favored or unfavored crystallization of L- and D-pHpgpTs. The primary needle-like molecular crystals before assembly should present optical activity in the CD spectrum despite their achiral morphology. Recently, Jiang et al. observed that second-order assembly displays opposite chirality from their primary building blocks (Science 2020, 368, 642-648). Therefore, I am curious about the chirality of the needle-like building blocks here, which should be identified using CD spectrum. Besides, how would the primary chiral crystal and second-order assembly contribute to their CD spectra of the final product?

5. Typically, oriented attachment (OA) describes the process in which inorganic crystalline nanoparticles align their atomic lattices and grow together into a single crystal. In this way, OA should involve processes of nanocrystal assembly, atomic alignment, and unification by attachment. Two aligned crystals become one larger crystal by epitaxy of two specific facets, one of each crystal. The authors claimed that the hierarchical crystal aggregates forms through OA process. However, no direct evidences on the lattice orientation are presented except for the energetic descriptions. The authors are suggested to take care of this growth concept.

Here are the point-to-point response:

Reviewer 1: This manuscript describes the isolation-crystallization of one enantiomer from a racemic mixture using a polymeric material that contain side groups of one of the enantiomers. The main finding of this article is that the amount added and the molecular weight of the polymer dictate the crystallization of one of the enantiomers. Thus one polymer additive can provide the separation of a racemic mixture by inducing crystal formation of one enantiomer.

This is a nice and detailed work that demonstrate one specific system tailored for the separation of a certain racemic mixture. It seems that for each racemic mixture, a polymeric version should be tailored and investigated to obtain selective crystallization.

The process described is not trivial and requires tailoring polymers of certain structure and molecular weights that may fit one set of racemic mixture and does not provide a general process where one is adding a magic molecule that fits many racemic mixtures, this would justify special attention. This work is specific and not of a general solution but a finding that is of interest for experts in the isolation procedures. Moreover, recent developments in the isolation of racemic mixtures using novel methods have been recently published.

Inducing crystallization of one enantiomer by adding a chiral agent is well known, this work is within this scope and provides a farther example of these efforts. This work is not a break through that justify publication in this journal. It should be directed to a specific journal that deal with racemic separations.

More specifically, this article is hard to read, the illustrations are difficult to follow as well as the details of the procedures.

In summary, this article is not recommended for publication in this journal, it should be forwarded to a specific journal in this field.

Thanks for your patient review to our manuscript. And your valuable comments will certainly help us improve our paper. We would like to clarify and highlight the novelty and the most notable merits of our work here and in the manuscript (lines 19-22).

First of all, a brand-new strategy has been developed in this work to tune chirality of hierarchically ordered organic crystals derived directly from a racemic solution by synergic combination of stereoselective and non-stereoselective interactions. Stereochemistry is originated from the famous work of L. Pasteur in 1848, who separated the enantiomeric crystals of tartrate. Since then, stereochemistry has rooted into all fields of chemical research. Generally speaking, a pair of enantiomers have identical chemical and physical properties, except in chiral environments. This is also the foundation of chiral resolution: a chiral selector must be offered (i.e., chiral stationary phase, chiral additives, chiral solvents, etc) to discriminate the enantiomers. We can regard chiral resolution as a special chemical reaction, but the reactants are usually not covalently bounded. The ideal model is that the chiral selector (s') can only interact with one of the enantiomers owing to precise stereochemical fit (Fig. R1a). But in fact, the selectors can interact with both enantiomers with different rate constants (k_S and k_R , Fig. R1b). For a long time, people have dedicated to improve one of the rate constants (i.e., stereoselective interactions) and suppress the other (i.e., non-

stereoselective interactions). Herein, we have developed a novel strategy that offers not only the preferred formation of one enantiomorph from racemic solution but also the subsequent enantiomer-specific oriented attachment of this enantiomorph by balancing stereoselective and non-stereoselective interactions (Fig. R1c), achieved by employing chiral polymeric additives with varying molar masses. The accessibility of enantiopure macroscopic organic crystals from racemic starting materials without further manual purification should be of great significance for chirality-related applications.

Figure R1. The mechanisms for chiral discrimination. (a) The ideal model for the chiral resolution aided by a chiral selector (s'), one enantiomer can form complex with s' , while the other cannot. (b) Actually, a selector usually has distinct interactions with both enantiomers. (c) In this work, stereoselective and non-stereoselective interactions are balanced to achieve hierarchical chirality control.

Even as a technique for chiral resolution, the findings developed in this work have advantages over the state of art methods. A chiral and non-racemic inducer is usually required in efficient crystallization-based resolutions, such as Viedma ripening, “tailor-made” additives, and those tuned by chiral nano-particles. One chiral inducer leads to the formation of one enantiomorph. The enantiomorph with opposite configuration have to be obtained by changing the stereostructure of inducer. Herein, for the first time, we report the preparation of enantiomorphs with desired configuration by changing the molar mass of polymeric additive but not its stereostructure. Thanks to the rapid development of living polymerization, the molar mass control of polymers is much easier than tailoring their structures. It will not only improve the efficacy of crystallization-based resolution but also deepen our fundamental understanding in accurate chiral discrimination and cross-scale, multi-level chiral transmission.

We agree with the reviewer on the point that a polymeric version, according to the “Rule of Reversal” (*J. Am. Chem. Soc.* **1982**, 104, 4610), should be tailored and investigated to obtain stereoselective crystallization for a group of racemates. One chiral additive leads to the formation of one enantiomorph. The enantiomorph with opposite configuration have to be obtained by changing the configuration of the additives. We report, for the first time, the preparation of enantiomorphs with desired configuration by changing the molar mass of polymeric additive but not its structure.

Reviewer 2: In this manuscript, Ye et al. reported a facile and effective route to achieve the enantioselective crystallization and assembly of hierarchically ordered chiral crystals from racemic molecules by introducing polymeric additives. By regulating the molar masses and amount of polymeric additives, the chirality of the molecular crystal aggregates can be modulated, yielding

either P- or M-type fan-shaped crystal aggregates of controllable ee% values. In particular, they reported enantiopure second-order assembly using polymer of P5. The accessibility of enantiopure macroscopic organic crystals from racemic starting materials without further manual purification is of great significance for chirality-related applications. Furthermore, combining theoretical calculations and experimental results, the authors also revealed the underlying mechanism that enables such chirality tenability. Overall, this work is interesting and can be considered for publication in Nature Communications after following comments are fully addressed.

We are grateful to the reviewer for the positive comments about our work.

1. The twisting direction of the resulting crystal aggregates is relatively difficult to resolve, the authors may provide eye-guided SEM images of higher resolution.

Figure R3. The images of fan-shaped crystals. (a) Picture of a left-handed fan. (b-c) The typical SEM images of left-handed (P-type) fan-shaped crystals of *D-pHpgpTs*. (d) Picture of a right-handed fan. (e-f) The typical SEM images of right-handed (M-type) fan-shaped crystals of *L-pHpgpTs*.

Thank you for your valuable suggestions. The eye-guided SEM images were re-recorded on a Hitachi S-4800 field emission scanning electron microscope operated at 10 keV, and some of the photos in Fig. 2b-e have been replaced and rearranged. The fan-shaped crystals with P- and M-type chirality were shown in Fig. R3, in which the layered crystalline building blocks and the tiny twist angle between two layers can be observed. For better comprehension, photos of folding fans with different opening directions were given (Fig. R3a,d).

2. The numbers of samples counted in Supplementary Figure 14 for statistical ee% values seem far from enough. Would the circular dichroism spectrum manifest their enantioselectivity differences in Figure 3(a)? The authors are encouraged to present CD spectra or g-factor spectra of these crystal aggregates to evaluate the ensemble chirality differences using different polymeric additives.

Figure R4. The CD and adsorption spectra of the DSB labeled crystals. (a) CD (up) and adsorption (down) spectra of fan-shaped crystal aggregates of *L*-pHpgpTs when 1.5 wt% P2 was used as the additive. (b) CD (up) and adsorption (down) spectra of fan-shaped crystal aggregates of *D*-pHpgpTs when 1.5 wt% P5 was used as the additive. (c) CD (up) and adsorption (down) spectra of fan-shaped crystal aggregates of pHpgpTs when 1.5 wt% P9 was used as the additive.

In our experiments, a 5 mL bottle was used as a crystallizer, and only 50-80 mg of crystals were obtained for each time. After the crystallization, around ten to twenty crystals were obtained. The experiments were repeated 2-4 times and all the resultant crystals were collected and counted in each case. Although the numbers of samples counted in Supplementary Figure 14 are indeed not big enough, they are based on all the crystals formed.

Although the CD tests are quite challenging, we have tried our best to do the experiments. The circular dichroism spectra of fan-shaped crystals were recorded by using diffused reflection light from the samples received through integrating sphere. By using this method, the cutoff wavelength should be greater than 250 nm, but the absorption of pHpgpTs is around 240 nm. In order to solve this problem, we run the crystallization by adding disodium sulphonated bathophenanthroline (DSB) to replace 5 mol% of the *p*-toluenesulfonic acid. The obtained red crystals were used to test the CD signals. The DSB were used as probe to detect the chirality. The crystal samples were fixed on the sample holder aided by double-sided tape. But the tape itself will greatly interfere the CD signals over 500 nm. We had to subtract the background absorption of the tape and test three times at different zones to collect the average data (Fig. R4). When the short chain length polymer (P2) was used as the additive, the crystals displayed two opposite bisignated signals, a negative signal at 315 nm and a positive one at 390 nm (Fig. R4a). When the middle chain length polymers (P5) was used

as the additive, the crystals also displayed cotton effect at the same wavelength but with opposite signals (Fig. R4b,d). When a long chain length polymer (P9) was used as the additive, the crystals displayed no obvious CD signals at the absorption range. These results were supplemented into the main text (lines 173-175).

3. As described in the manuscript, the synergistic combination of stereoselective and non-stereoselective interactions of the polymeric additives results in favored or unfavored crystallization of L- and D-pHpgpTs. The primary needle-like molecular crystals before assembly should present optical activity in the CD spectrum despite their achiral morphology. Recently, Jiang et al. observed that second-order assembly displays opposite chirality from their primary building blocks (Science 2020, 368, 642-648). Therefore, I am curious about the chirality of the needle-like building blocks here, which should be identified using CD spectrum. Besides, how would the primary chiral crystal and second-order assembly contribute to their CD spectra of the final product?

Figure R5. The evolution of the DSB labeled crystals. (a) CD (up) and adsorption (down) spectra of the pHPgpTs solution with 1.5 wt% P5 as additive at different temperature. (b) The correlation between temperature and CD signals at 578 nm. (c) CD (up) and adsorption (down) spectra of the fan-shaped crystals obtained at 5 h and 24 h. (d) The typical SEM images of the obtained crystals at the first stage and the second stage. (e) The typical SEM images of the obtained crystals at the second stage and the third stage.

We appreciate these advices. N. A. Kotov et al. reported the second-order assembly of Au nanoparticles to form chiral crystal aggregates (Science 2020, 368, 642-648, ref. 23 in the previous version of our manuscript). In that work, Au particles have strong CD signals in visible light range

and is suitable for CD test. Besides, these nanoparticles won't redissolve once generated, which can be diluted to suitable concentration for the CD test. In our case, the concentration for crystallization is far surpassed the detection limit and the formed crystals will be redissolved when the aqueous solution was diluted. More importantly, the short-wavelength CD signals of the amino acids is quite weak.

Nevertheless, we have tried our best to collect the CD signals of samples at different stages by combining two modes of CD test. A slightly diluted solution was prepared, and the temperature was gradually dropped down to 10 °C to trigger the crystallization. 1.5 wt% P5 was used as the additive to generate fan-shaped *D*-crystals. To avoid the absorption zone of *pHpgpTs*, which is far surpassed the detection limit, all the samples were labeled by DSB for the detection of chirality.

Just like the evolution process described in the main text. The cooling crystallization process still showed three distinct stages. Stage I, from 50-25 °C, the whole solution turned from transparent to slightly turbid and no obvious precipitate could be observed. At this stage nano-scale particles were dominated (Fig. R5d, left). When the solution was kept at 10 °C for 1.5-5 h (stage II), the solution became more turbid and precipitation took place at the bottom of the bottle. At this stage, more tiny crystalline platelets appeared and the stacking of two or three platelets was also present (Fig. R5d, right and Fig. R5e, left). When the solution was kept 10 °C for more than 5 h (stage III), the supernatant became clear and fan-shaped crystals gathered at the bottom (Fig. R5e, right).

For the samples at stage I and II, the CD signals were recorded by receiving the transmitted light. It displayed two opposite bisignated signals in the visible region, a negative signal at 478 nm and a positive one at 578 nm (Fig. R5a). The relationship between CD signal at 578 nm and temperature was plotted (Fig. R5b). From 50 to 25 °C, the CD signals gradually increased, which indicated the formation of primary nano-scaled building blocks. It should be noted that very weak CD signals can still be detected even for the starting solution, this may be due to the induction of chiral amino acid molecules. From 25 to 10 °C, no complex structures generated and the CD signal kept steady. When the sample was kept at 10 °C for 90 min, the initial assembly of two or three needle-like crystals occurred and the CD signals increased again (Fig. R5d).

We collected the crystals at 5 h and 24h, and tested their CD signals by using diffused reflection light from sample received through integrating sphere. It is worth noting that there is no comparability between the CD signals of solutions and solids. The crystals obtained at 5 h had a broad positive signal below 350 nm, while the crystals obtained at 24 h showed obvious cotton effects at the same wavelength (Fig. R5c). The cotton effects indicated the emergence of high-level chirality and better arrangement of chromophores (Fig. R5e). These results were supplemented into the main text (lines 140, 143 and 148).

4. As shown in Figure 3, both the Mn and adding amount of polymeric additives show influences on the chirality of hierarchical assembly. Thus, is it possible to achieve both enantiopure (ee value of 100%) P- and M-type assembly by synergistically modulating Mn and adding amount of polymeric additives?

Figure R6. The HPLC results of the *pHpgTs* crystals. (a) The M-type fan-shaped crystals. (b) needle-like crystals.

When using the short chain polymers as additives, the ee% values of the obtained crystals was relatively small. It should be noted that this ee% value was for the whole crystal mixtures. When using short chain polymers as additives ($M_n < 11000$), both needle-like crystals and fan-shaped crystals were obtained, and this was inevitable even though we changed the adding amount and molecular weight. We separated the two kinds of crystals by manual selection and did the HPLC test for both of them. It showed that all of the needle-like crystals were *D*-crystals (Fig. R6b). While the ee% value for the fan-shaped crystals was as high as 70% (*L*-crystals) (Fig. R6a). These results were added into supplementary information (Supplementary Figure 16)

5. Typically, oriented attachment (OA) describes the process in which inorganic crystalline nanoparticles align their atomic lattices and grow together into a single crystal. In this way, OA should involve processes of nanocrystal assembly, atomic alignment, and unification by attachment. Two aligned crystals become one larger crystal by epitaxy of two specific facets, one of each crystal. The authors claimed that the hierarchical crystal aggregates forms through OA process. However, no direct evidences on the lattice orientation are presented except for the energetic descriptions. The authors are suggested to take care of this growth concept.

Figure R7. Crystal aggregates reported in literatures. (a) SEM image of *DL*-alanine mesocrystals (scale bar=100 μ m). (b) The view of one of the rougher faces revealing the inner mesostructure (scale bar=2 μ m) (Chem. Eur. J. 2005, 11, 2903). (c-d) Polarized-light micrographs with the analyzer rotated to darken l-crystals (left), then d-crystals (right) (Angew. Chem. Int. Ed. 2013, 52, 10545).

We agree with you on the point that oriented attachment (OA) describes the process in which crystalline nanoparticles align their atomic lattices and grow together into a single crystal. We have no direct evidence on the lattice orientation and the direct observation of the lattice of organic crystal is a tough challenge. H. Cölfen et al used this concept on orderly arranged organic crystal aggregates (Chem. Eur. J. 2005, 11, 2903). In his work, mesocrystals of *DL*-alanine were reported (Fig. R7a). Higher resolution SEM pictures of the rough sites of less perfect species reveal the inner alignment of the platelet-like crystals within these superstructures (Fig. R7b). Although the whole crystal does not show any chiral morphology, the alignment of the building blocks is similar to our work. Besides, C. Viedma and coworkers raised the concept of “enantiomer-specific oriented attachment (ESOA)” (Angew. Chem. Int. Ed. 2013, 52, 10545) to describe the process of millimeter-scale crystals to form large, homochiral aggregates. Homochiral aggregates of macroscopic NaClO_3 crystals were formed by ESOA and examined by polarized-light microscope (Fig. R7c,d). In our case, the final fan-shaped crystals were gradually assembled by the platelet sub-structures that formed in the earlier stages. The staggered crystalline layers were similar to that in Cölfen’s work. And the phenomenon of the stereoselective aggregation was quite similar to Viedm’s work. For these reasons, we prefer to follow this description.

REVIEWERS' COMMENTS

Reviewer #2 (Remarks to the Author):

The authors have well answered my questions and made suitable revision. This revised manuscript can be accepted as is.